# Preparation and Properties of Ce_0.8_Sm_0.16_Y_0.03_Gd_0.01_O_1.9_-BaIn_0.3_Ti_0.7_O_2.85_ Composite Electrolyte

**DOI:** 10.3390/ma15165591

**Published:** 2022-08-15

**Authors:** Yajun Wang, Changan Tian, Minzheng Zhu, Jie Yang, Xiaoling Qu, Cao Chen, Cao Wang, Yang Liu

**Affiliations:** 1College of Chemistry and Civil Engineering, Shaoguan University, Shaoguan 512005, China; 2College of Energy Material and Chemical Engineering, Hefei University, Hefei 230601, China

**Keywords:** composite electrolyte, conductivity, sol-gel method, solid-state reaction

## Abstract

Samarium, gadolinium, and yttrium co-doped ceria (Ce_0.8_Sm_0.16_Y_0.03_Gd_0.01_O_1.9_, CSYG) and BaIn_0.3_Ti_0.7_O_2.85_ (BIT07) powders were prepared by sol-gel and solid-state reaction methods, respectively. CSYG-BIT07 composite materials were obtained by mechanically mixing the two powders in different ratios and calcining at 1300 °C for 5 h. Samples were characterized by X-ray diffraction (XRD), scanning electron microscopy (SEM), as well as electrical properties and thermal expansion coefficient (TEC) measurements. A series of CSYG-BIT07 composite materials with relative densities higher than 95% were fabricated by sintering at 1300 °C for 5 h. The performance of the CSYG-BIT07 composite electrolyte was found to be related to the content of BIT07. The CSYG-15% BIT07 composite exhibited high oxide ion conductivity (*σ*_800°C_ = 0.0126 S·cm^−1^ at 800 °C), moderate thermal expansion (TEC = 9.13 × 10^−6^/K between room temperature and 800 °C), and low electrical activation energy (*E*_a_ = 0.89 eV). These preliminary results indicate that the CSYG-BIT07 material is a promising electrolyte for intermediate-temperature solid oxide fuel cells (IT-SOFCs).

## 1. Introduction

Solid oxide fuel cells (SOFCs) have attracted considerable attention in recent years, owing to their high efficiency, environmentally friendly nature, fuel flexibility, and low cost [1,2,3]. SOFCs are composed of a porous oxide-based cathode, a dense ceramic electrolyte, and a porous ceramic–metal composite anode. Among these components, the electrolyte plays a key role in conducting ions, isolating fuels, and oxidizing gases. Nowadays, 8 mol% yttrium-stabilized zirconia (YSZ) is widely used as electrolyte material in SOFCs, owing to its pure oxygen ion conductivity and good compatibility with electrode materials [4,5]. YSZ needs to be operated at high temperature (800–1000 °C) to attain the required conductivity and high power output. This high working temperature poses a great challenge for the engineering of the various components of the SOFCs. Therefore, reducing the operating temperature has become a crucial task, which requires developing alternative electrolytes with higher ionic conductivity than YSZ [6,7].

Compared to YSZ, doped ceria is considered as one of the most promising electrolytes for IT-SOFCs. Doped ceria has many advantages, such as high oxide ion conductivity, high thermal stability, good compatibility with the electrodes, and good performance at low temperature [8,9,10,11]. When CeO_2_ is doped with divalent alkaline earth or trivalent rare earth metal oxides, the concentration of oxygen vacancies increases, due to the charge compensation mechanism, and thus the ionic conductivity is improved. However, CeO_2_-based materials tend to be partially reduced to Ce^3+^ in a reducing atmosphere at high temperature, cause electronic conductivity that will lead to a rapid degradation of the cell performance, and limit the extensive application of CeO_2_-based materials as electrolytes in SOFCs [9,10,11,12]. Co-doping approaches have been introduced to reduce the electronic conductivity. Previous studies found that ceria doped with two or more cations showed significantly higher ionic conductivity than the single-doped ceria electrolytes. Co-doping suppresses the ordering of oxygen vacancies and the reduction of Ce^4+^; this reduces the activation energy, and thus increases the ionic conductivity. Ce_0.85_(Sm_x_Nd_1−x_)_0.15_O_2−δ_, Ce_0.80_Gd_0.2−x_Pr_x_O_1.9_, Ce_0.80_Gd_0.2−x_Sm_x_O_1.9_, Ce_0.82_Sm_0.16_Sr_0.02_O_1.90_, Ce_0.8_Cu_x_Mn_0.1−x_Zr_0.1_O_2,_ Ce_0.82_La_0.06_Sm_0.06_Gd_0.06_O_2−δ_, and Ce_0.85_Gd_0.10_Mg_0.05_O_2−δ_ are some well-known examples [3,9,12].

On the other hand, Ba_2_In_2_O_5_-based materials can serve as alternative electrolytes for IT-SOFCs, owing to their relatively high ionic conductivity and good compatibility with the cathode materials. Ba_2_In_2_O_5_ undergoes a phase transition from a high-temperature phase with perovskite structure to a low-temperature phase with brownmillerite structure at around 925 °C. The phase transition causes a sudden drop in conductivity below 925 °C. Previous studies have shown that substitution of In by Ti in the brownmillerite compound Ba_2_In_2_O_5_ induces disorder in the ordered arrangement of oxygen vacancies. At higher substitution levels (>15%), the compound transforms into a disordered perovskite structure at room temperature, exhibiting better anionic conductivity than the parent Ba_2_In_2_O_5_ phase [13,14,15,16,17,18]. In this context, the perovskite-type oxide BaIn_0.3_Ti_0.7_O_2.85_ (BIT07) is an oxygen ion conductor with a conductivity similar to YSZ. Moreover, BIT07 presents an excellent structural compatibility with perovskite cathode substrates [13,14,16,17,18], and has proven to be a suitable electrolyte material for SOFCs.

In recent years, composite electrolytes consisting of at least two phases, i.e., a doped ceria main phase and other phases, have been successfully developed [19,20,21,22,23,24]. This double-matrix structure (phase) can overcome the drawbacks of single-phase materials and retain their advantages. The composite can effectively suppress the transformation from Ce^4+^ to Ce^3+^ of CeO_2_-based materials, and exhibits high ionic conductivity, high efficiency, and improved fuel cell performance at lower temperatures [12,19,20]. Therefore, combining the co-doping and composite formation approaches may further enhance the performances of SOFCs.

Composite electrolytes consisting of doped ceria and BIT07 have rarely been investigated. In this study, Sm^3+^, Gd^3+^, and Y^3+^ triple-doped CeO_2_-BaIn_0.3_Ti_0.7_O_2.85_ (CSYG–BIT07) composite electrolytes were prepared to further enhance the ionic conductivity at intermediate/low temperatures. In addition, we investigated the effect of the BIT07 content on the phase composition, structure, and electrical conductivity of CSYG-BIT07 composite electrolytes.

## 2. Experimental

### 2.1. Sample Preparation

The Ce_0.8_Sm_0.16_Y_0.03_Gd_0.01_O_1.9_ (CSYG) material was synthesized via the sol-gel method. Sm(NO_3_)_3_·6H_2_O (99.9%), Y(NO_3_)_3_·6H_2_O (99%), Gd(NO_3_)_3_·6H_2_O (99.9%), Ce(NO_3_)_3_·6H_2_O (99.9%), and citric acid (99.9%) were dissolved separately in deionized water. Then, the solutions were mixed with a 3:2 molar ratio between citric acid and metallic ions. To promote the complexation of the metal ions, the pH was adjusted to around 7 by slow addition of ammonia. The as-mixed solution was stirred in a water bath maintained at a constant temperature of 80 °C for 4 h to form the sol. Upon water evaporation, a gel was obtained. The gel was poured into a crucible and calcined in a muffle furnace to generate loose powders, which were held at 600 °C for 3 h to obtain the initial CSYG powders. The synthesis procedure is shown in Figure 1.

BIT07 compounds were prepared by solid-state reaction of Ba(NO_3_)_2_ (99.9%), In_2_O_3_ (99.9%), and TiO_2_ (99%). Ba(NO_3_)_2_, In_2_O_3_, and TiO_2_ were weighed in appropriate amounts and mixed with ethanol in an agate mortar. The mixed oxides were compacted into a pellet and heated to 750 °C for 3 h, followed by cooling to room temperature. Finally, the resulting products were ground into finer powders using an agate mortar.

To prepare the composite electrolytes, the resulting CSYG and BIT07 powders were mixed with different molar ratios (5, 10, 15, 20, and 25 mol%) of BIT07, and ball-milled with zirconia balls for 24 h to obtain a uniformly mixed composite powder. After drying, an evenly mixed CSYG-BIT07 composite powder was obtained. According to the amount of BIT07 in the composite material, the obtained CSYG-BIT07 composite electrolytes were denoted as 5%BIT07, 10%BIT07, 15%BIT07, 20%BIT07, and 25%BIT07.

The as-synthesized CSYG-BIT07 powders were pressed at 10 MPa to obtain pellets of 12 mm in diameter and ~1.0 mm in thickness. Green pellets were sintered at 1300 °C for 5 h with a heating rate of 5 °C/min in air atmosphere, and then allowed to cool to room temperature. The preparation procedure of CSYG-BIT07 is shown in Figure 2.

### 2.2. Characterizations

Phase analysis of the samples was carried out by X-ray diffraction (XRD, Model Rigaku) with a 2*θ* range of 20–80°, a scanning rate of 5°/min, and a Cu K_α_ radiation source (λ = 0.15406 nm). Phase identification was performed using Joint Committee on powder diffraction standards (JCPDS) data and the Jade evaluation software. The surface morphology of the electrolyte was inspected by scanning electron microscopy (SEM, Model SU8220). To evaluate electrical performances, both sides of the sintered pellets were coated with silver paste and dried at 700 °C for 15 min to remove organic matter. Impedance measurements of the pellets were carried by two-probe impedance spectroscopy from 300 to 800 °C at intervals of 50 °C, in the frequency range of 10^5^–10^−1^ Hz, using a GSL-1100X AC impedance analyzer. Data were collected and fitted to an equivalent circuit using the Zview software.

A 8 × 4 × 4 mm^3^ sintered sample was prepared to perform thermal expansion measurements from room temperature to 1000 °C at a heating rate of 5 °C/min in air, using a dilatometer (NETZSCH DIL 402 PC). The degree of thermal expansion of a material can be expressed in terms of the linear expansion coefficient α.

CSYG-BIT07 electrolyte (thickness: 0.4 mm)-supported electrodes and single cells were fabricated by printing. The anode slurry was prepared by mixing 40 wt% CSYG-BIT07 and 60 wt% NiO powders with a binder consisting of ethyl cellulose and terpineol. The cathode slurry was prepared by mixing 50 wt% CSYG-BIT07 and 50 wt% LSCF(La_0.6_Sr_0.4_Co_0.2_Fe_0.8_O_3−δ_) powders with the same binder used for the anode slurry. The mixture was printed on the electrolyte as a circle with 6 mm diameter. After printing the electrode slurries onto the sintered electrolyte disk, the cell was burned at 550 °C for 4 h and then sintered in air at 1100 °C for 2 h. This process was repeated two times to increase the electrode thickness. Electrical performance measurements were conducted with an electrochemical analyzer (CHI-614D CHI, China). 

## 3. Results and Discussion

### 3.1. Structural Analysis

Figure 3a displays the powder XRD patterns of CSYG and BIT07 at room temperature. The figure shows that the CSYG and BIT07 samples exhibited pure fluorite (JCPDS card no. 34-0394) and perovskite structures [17,18,19], respectively. The pattern of BIT07 showed a single perovskite phase, implying that Ti atoms were fully incorporated into the perovskite lattice, suppressing phase transitions and maintaining the perovskite phase at room temperature. This is due to the fact that the substitution of Ti for In produces oxygen vacancies in the Ba_2_In_2_O_5_ lattice; these vacancies can suppress the distortion to the orthorhombic symmetry and inhibit crystal phase transitions [16]. 

The XRD patterns of the CSYG-BIT07 composite electrolytes sintered at 1300 °C for 5 h are shown in Figure 3b. Two different phases were identified in the pattern of 5%BIT07, corresponding to CSYG and BIT07. New peaks were detected with increasing content of BIT07, corresponding to impurity phases formed during the sintering of CSYG and BIT07 under the selected conditions. The new other phases can be confirmed as BaTiO_3_ and Ba_2_In_2_O_5_. These results suggest the occurrence of a chemical reaction between the CSYG and BIT07 components and this is a serious problem which affects the use of CSYG-BIT07 in SOFCs. Further inspection of the 2*θ* region around 28.5° (Figure 3) reveals a shift in the (111)_CSYG_ diffraction peak of the CSYG-BIT07 composite electrolyte relative to the pure CSYG phase. According to Bragg’s law, a shift in a diffraction peak is usually related to a variation in the lattice parameters. The lattice parameter (a_0_) of the cubic fluorite structure was calculated and shown in Table 1. In addition, it is worth noting that the degree of peak shift in the CSYG-15%BIT07 composite is greater than the composite with the other amount of BIT07. As a result, the unit cell volume is increased, which can be ascribed to the pinning effect of the new phase (i.e., BaTiO_3_ or Ba_2_In_2_O_5_). The slight diffusion of In^3+^ (0.80 Å) atoms is beneficial for increasing the number of oxygen vacancies in CSYG (the defect equation is provided below), which may result in an increase in ionic conductivity.
(1)$${\mathrm{In}}_{2}O_{3}\overset{2{\mathrm{CeO}}_{2}}{\to }2{\mathrm{In}}_{\mathrm{Ce}}^{\prime }+3O_{O}^{\times }+V_{O}^{••}$$

### 3.2. Density and Microstructure

The relative density of CSYG-BIT07 sintered in air at 1300 °C for 5 h was measured using the Archimedes’ principle, and is shown in Table 1. It can be seen that the relative density of all samples is greater than the theoretical value by 95%. Moreover, the relative density increased when BIT07 was added, and then decreased with further increases in BIT07 content. The CSYG-15%BIT07 sample sintered in air at 1300 °C for 5 h exhibited the highest relative density of 98.8%, indicating that the densification temperature of CSYG-BIT07 was 100–200 °C lower than the CeO_2_-based materials prepared by solid-state synthesis [25,26,27]. The CSYG-BIT07 samples showed low temperature sinterability, which may assist the co-firing of other cell components of IT-SOFCs at reduced temperatures [28,29].

Figure 4 shows the cross-sectional SEM images of CSYG-15%BIT07 sintered at 1300 °C for 5 h. The CSYG-15%BIT07 sample consisted of well-defined grains separated by grain boundaries, and exhibited a sufficiently dense structure with few pores at grain boundaries or at the triple junction point. The relative density and SEM analysis results show that the composite electrolyte had a high density and could thus meet the corresponding requirements.

### 3.3. Impedance Analysis

The AC impedance method is an effective approach to study the electrical properties of solid electrolytes. The electrical properties of CSYG-BIT07 were studied by two-probe impedance spectroscopy. A typical impedance spectrum of the SOFC electrolyte consisted of three semicircles at high, intermediate, and low frequencies, corresponding to the grain (*R*_b_), grain boundary (*R*_gb_), and electrode (*R*_ct_) resistances, respectively [30,31]. Figure 5 shows the AC impedance spectra and equivalent circuit model of the CSYG-15%BIT07 sample sintered at 1300 °C, measured at different temperatures in air.

Figure 5a displays complex impedance graphs (Nyquist plots) of the samples at 500 °C. The complex impedance plots were fitted by the Zview software with the typical equivalent circuit shown in Figure 6a, which consisted of three parallel resistances (*R*) and constant phase elements (CPEs) connected in series. At high temperatures, the impedance decreased and the semicircles were shifted toward higher frequencies (Figure 5b–d). Increasing the temperature results in increased lattice thermal motion of the electrolyte material. Owing to non-Debye relaxation phenomena, the impedance arcs shift to higher frequencies with increasing temperature. This leads to the disappearance of the arcs associated with grains at higher temperatures, owing to the limited frequency range of the measurements [30,31,32,33].

The grain and grain boundary resistances were determined using equivalent electrical circuits. The total resistance (*R*_t_) of the sample was the sum of *R*_b_ and *R*_gb_. The total ionic conductivity (*σ*_t_) at different temperatures was determined using the following equation [28,29,30,31]:(2)$$\sigma_{t}=\frac{D}{S\cdot R_{t}}$$
where *D*, *S*, and *R*_t_ represent the thickness, area, and resistance of the sample, respectively.

Figure 6 shows that the total ionic conductivity of all samples increased with the temperature. Moreover, the conductivity of CSYG-BIT07 increased with the BIT07 content and reached a maximum for a content of 15%. Thereafter, the conductivity decreased with further increases in the BIT07 content. As shown in Table 1, the conductivities at 800 °C of CSYG-BIT07 samples with different BIT07 contents followed the order 15%BIT07 > 20%BIT07 > 25%BIT07 > 10%BIT07 > 5%BIT07, which indicates that BIT07 plays an important role in determining the conductivity. CSYG-15%BIT07 showed the highest conductivity of 0.0126 S·cm^−1^ at 800 °C.

Figure 7 shows an Arrhenius plot of the total conductivity of the CSYG-BIT07 electrolyte. The total conductivity data in the figure were well fitted with a straight line throughout the operating temperatures. This indicates that the temperature dependence of the total conductivity of the CSYG-BIT07 electrolyte can be expressed by the Arrhenius equation [31,32,33]:(3)$$\sigma_{t}=\frac{\sigma_{0}}{T}\exp(-\frac{E_{a}}{kT})$$
where *σ*_0_, *E*_a_, *k*, and *T* are the pre-exponential factor, activation energy, Boltzmann constant, and absolute temperature, respectively.

The activation energies calculated from the slope of the Arrhenius plot are shown in Table 1. The activation energies of CSYG-5%BIT07, CSYG-10%BIT07, CSYG-15%BIT07, CSYG-20%BIT07, and CSYG-25%BIT07 were 0.94, 0.99, 0.89, 0.92, and 1.01 eV, respectively. Therefore, we can conclude that the activation energy of the CSYG-BIT07 electrolyte depends on the amount of BIT07.

### 3.4. Thermal Expansion 

The thermal expansion coefficient (TEC) is an important property of SOFCs. The TECs of the electrolyte, cathode, and anode should be close to each other to avoid microcracks and interfacial stress at the operating temperature [32,33]. The average thermal expansion coefficient can be determined by the following equation [34,35]:(4)TEC=(L−L0)L0T−T0
where *L*_0_ is the original length (um) of the synthesized sample, *L* is the sample length (um) at the corresponding temperature, while *T*_0_ and *T* are the initial and final temperatures (K), respectively.

Figure 8 shows the thermal expansion curves of the CSYG-15%BIT07 electrolyte, which exhibited a linear trend in the temperature range of 400–1000 °C. The average value of the thermal expansion coefficient was calculated to be 9.13 × 10^−6^/K at temperatures ranging from 25 to 800 °C. The TEC values for cathode materials, such as La_0.7_Sr_0.3_MnO_3_ (LSM), La_0.6_Sr_0.4_Co_0.2_Fe_0.8_O_3−δ_ (LSCF), and La_0.7_Sr_0.3_MnO_3_–Zr_0.92_Y_0.08_O_1.92_ (LSM-YSZ) have been reported to be ~12 × 10^−6^, 14–15 × 10^−6^, and 10.5 × 10^−6^/K, respectively [34,35,36]. The TEC of Ni-Zr_0.92_Y_0.08_O_1.92_ (Ni-YSZ), Ni-Sm_0.2_Ce_0.8_O_3−δ_ (Ni-SDC), and strontium titanate anode materials have been reported to be 12.3 × 10^−6^, 13.5 × 10^−6^, and 10.8 × 10^−6^/K, respectively [34,37,38]. This indicates that the measured TEC value for CSYG-15%BIT07, obtained in the present work, is compatible with the TEC of this composite with LSM-YSZ and SrTiO_3_ based electrodes [31,36,39].

### 3.5. Single Cell Performance 

To evaluate the effect of the CSYG-15%BIT07 composite electrolyte on the output performance of SOFCs, we fabricated a single cell with NiO + CSYG-BIT07|CSYG-BIT07|LSCF + CSYG-BIT07 configuration. Figure 9 shows the current density vs. cell voltage (*I*–*V*) and current density vs. power density (*I*–*P*) curves of the CSYG-15%BIT07 electrolyte-based cell at 800 °C. The open circuit voltage (OCV) and current density only reached ~0.5 V and 0.32 A cm^−2^, respectively. The maximum power output was ~0.16 W cm^−2^ at 800 °C, which is lower than the published data [23,31,32] for Ce_0.9_Gd_0.1_O_1.95_-1mol% MgO (0.47 W cm^−2^ at 700 °C) [40], La_0.9_Sr_0.1_Ga_0.8_Mg_0.2_O_2.85_-Ce_0.8_Gd_0.2_O_1.9_ (60 W cm^−2^ at 700 °C) [22], Ce_0.8_Sm_0.15_Ca_0.05_O_1.875_ (0.170 W cm^−2^ at 700 °C) [34], and La_0.85_Sr_0.15_Ga_0.8_Mg_0.2_O_2.825_ (0.185 W cm^−2^ at 700 °C) [41]. The fuel cell performance is relatively poor, which will limit the use of CSYG-BIT07 in SOFCs. Therefore, extensive theoretical and experimental investigations are required to improve the fuel cell performance in future work.

## 4. Conclusions

In the present work, a new type of composite electrolyte, CSYG-BIT07, was developed and used as electrolyte for IT-SOFCs. We characterized the structure, sinterability, and thermal/electrical properties of CSYG-BIT07 to study the effect of the BIT07 content. The results reveal that the new composite material was a two-phase composite ceramic. This shows that doping with BIT07 can promote the densification of electrolyte materials and lower the sintering temperature. CSYG-BIT07 electrolytes with relative densities greater than 95% were prepared by ball-milling a mixture of CSYG and BIT07 powders and sintering at 1300 °C for 5 h. The performance of the composite depends on the content of BIT07. A maximum oxide ionic conductivity (*σ*_800°C_ = 0.0126 S·cm^−1^), minimum electrical activation energy (*E*_a_ = 0.89 eV), and moderate thermal expansion coefficient (TEC = 9.13 × 10^−6^/K between room temperature and 800 °C) were achieved by adding 15 mol% of BIT07. These results highlight the great application prospects of the CSYG-BIT07 composites in IT-SOFCs.

## Figures and Tables

**Figure 1 materials-15-05591-f001:**
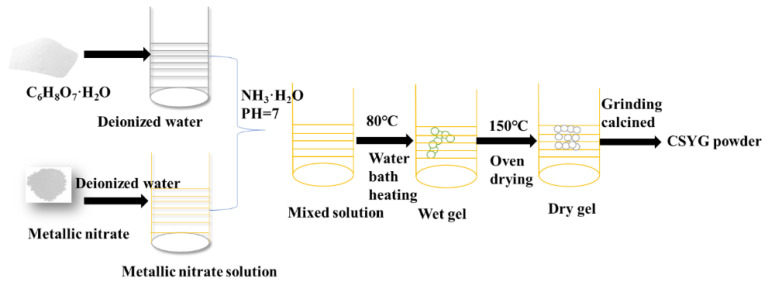
Flow chart of preparation of CSYG powders.

**Figure 2 materials-15-05591-f002:**
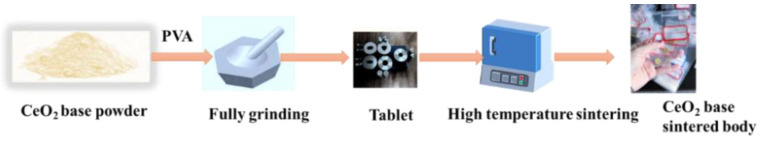
Flow chart of preparation of CSYG-BIT07.

**Figure 3 materials-15-05591-f003:**
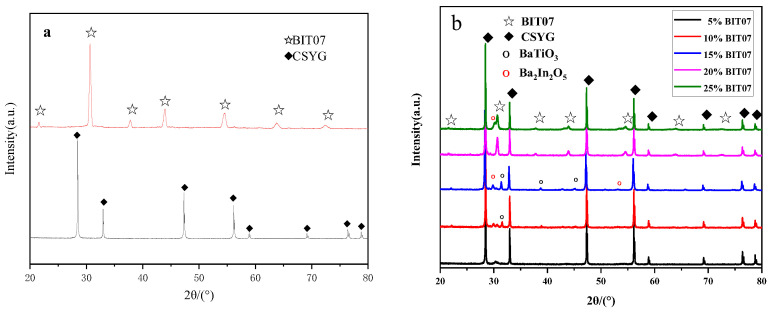
XRD pattern of (**a**) CSYG and BIT07, (**b**) CSYG-BIT07 composite materials.

**Figure 4 materials-15-05591-f004:**
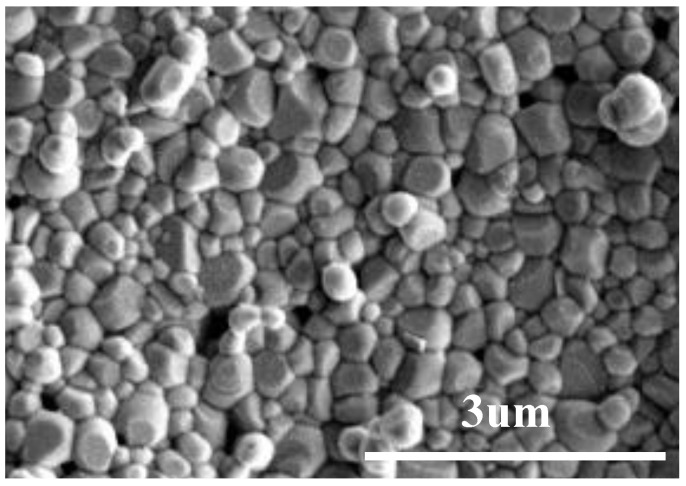
SEM images of CSYG-15%BIT07 sintered at 1300 °C for 5 h.

**Figure 5 materials-15-05591-f005:**
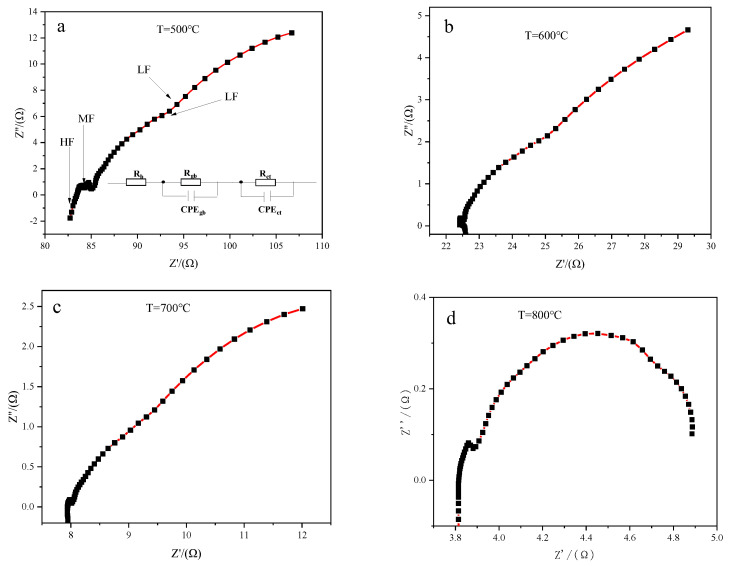
AC impedance graph of CSYG-15%BIT07 at different temperatures. (**a**) 500 °C, (**b**) 600 °C, (**c**) 700 °C and (**d**) 800 °C.

**Figure 6 materials-15-05591-f006:**
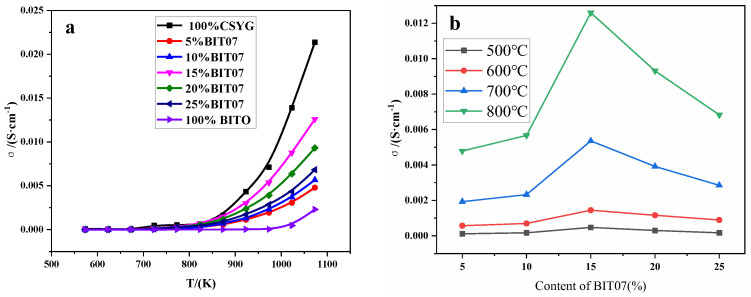
Ionic conductivity of CSYG-BIT07. (**a**) Ionic conductivity vs. temperature. (**b**) Ionic conductivity vs. content.

**Figure 7 materials-15-05591-f007:**
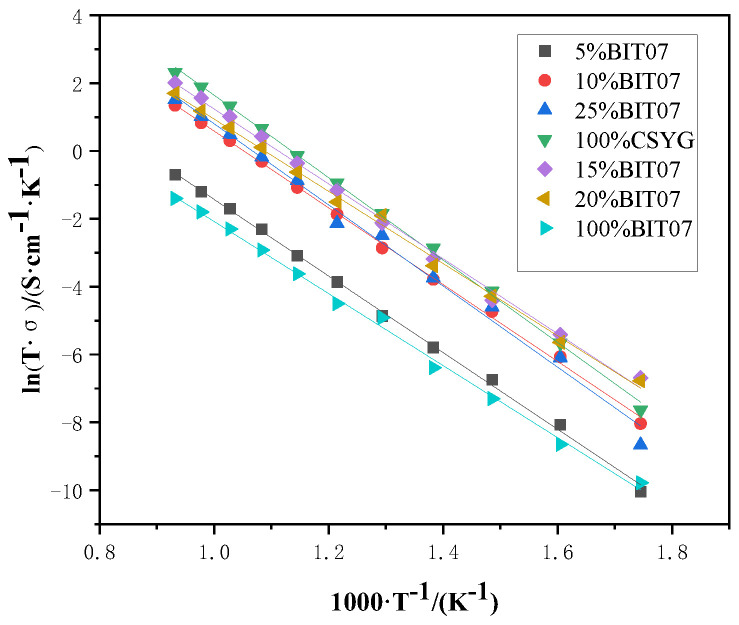
Arrhenius plots for total conductivity of CSYG-BIT07.

**Figure 8 materials-15-05591-f008:**
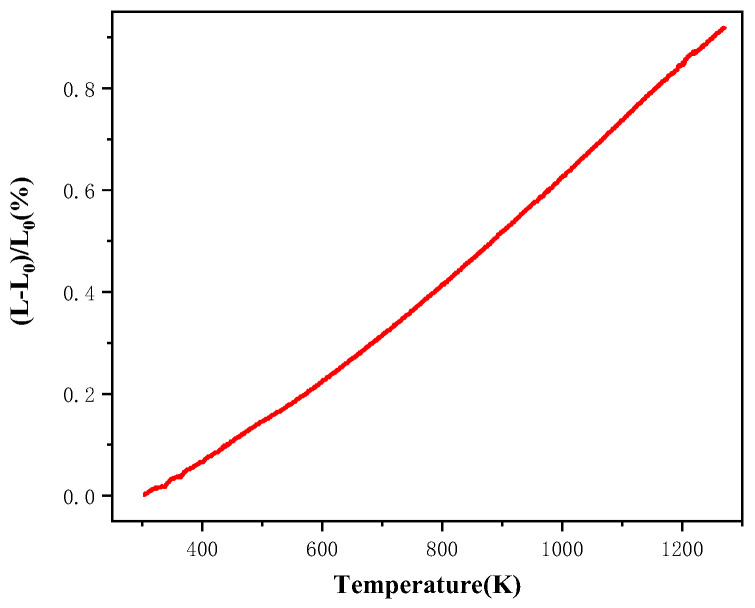
Thermal expansion curves of CSYG-15%BIT07.

**Figure 9 materials-15-05591-f009:**
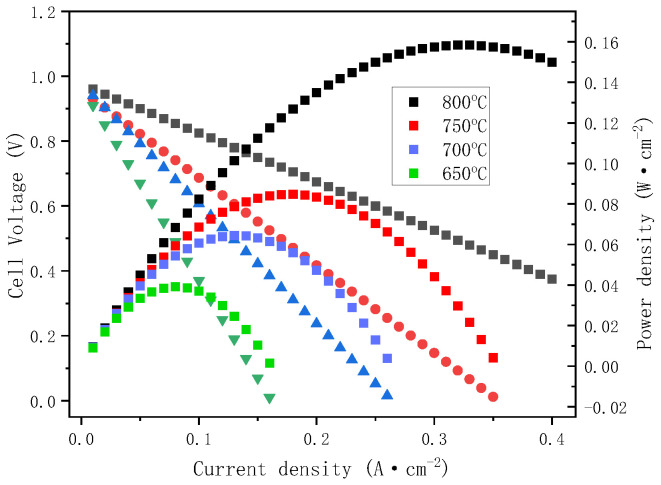
Single cell performance of a single IT-SOFC. NiO + CSYG-BIT07|LSCF + CSYG-BIT07 measured at 800 °C.

**Table 1 materials-15-05591-t001:** Properties of CSYG-BIT07 composite electrolyte.

Composition (BIT07 Content)	Relative Density/%	Lattice Parameter/(Å)	σ_700°C_/(mS·cm^−1^)	σ_800°C_/(mS·cm^−1^)	Ea/(eV)
5%BIT07	95.4	5.432	1.93	4.78	0.94
10%BIT07	95.6	5.436	2.32	5.67	0.99
15%BIT07	98.8	5.462	5.36	12.60	0.89
20%BIT07	96.7	5.443	3.92	9.31	0.92
25%BIT07	96.4	5.448	2.85	6.83	1.01

## Data Availability

Data is contained within the article.

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
