# Peer review of "Preparation and Properties of Ce0.8Sm0.16Y0.03Gd0.01O1.9-BaIn0.3Ti0.7O2.85 Composite Electrolyte"

_materials, 2022, doi:10.3390/ma15165591_

Round 1

Reviewer 1 Report

In this work, the Ce0.8Sm0.16Y0.03Gd0.01O1.9 and BaIn0.3Ti0.7O2.85 composites with various ratios have been prepared and evaluated as new electrolytes for solid oxide fuel cells. The crystal structure properties, Density and Microstructure, electrical conductivity, thermal expansion and electrochemical properties in SOFC are investigated. Unfortunately, the work cannot be published without significant improvements. For this work, extensive editing of English language and style is mandatory. It is hard to follow the work. The paper in present form in my opinion does not meet the requirement of the journal of Materials. However, the work can be resubmitted after significant improvements.

In the following, there are some suggestions for improvement which can be considered in a future submission.  

1. The introduction part should be optimized to justify the study.  

2. In the experimental part, the characterization methods and all used apparatus should be described in details, such as: SEM instrument, SOFC construction procedure and measurements. The composite ratio for electrodes and chemicals applied should be described. The electrode working area and the thickness of electrolyte should be presented.

3. Please provide separate XRD patterns for pure CSYG and pure BIT07 materials.

4. IR spectra data of pure CSYG and pure BIT07 should be included.

5. Captions for Figure 5 a-d should be specified.

6. In Figure 6, the fitting curve should be given.

7. The conductivity data of pure CSYG and pure BIT07 should be included for comparison.

8. For the thermal expansion coefficient, the unit should be K-1 not C-1.

9. In the section of “3.5. Single Cell Performance”, at first sight, the cell performance seems to be excellent. However, according to the V-I curve, the power density should be significantly lower. For instance, at 800C, the maximum power output can be archived at around 0.5 V with 0.3 Acm-2, generating around 0.15 W cm-2. In addition, scales on the horizontal and vertical axes should be kept in Figure 10.

Reviewer 2 Report

The manuscript deals with the preparation and characterization of various ceramic composites obtained by mixing RE-doped ceria and Ba-In-Ti perovskite in different ratios.

The work is introduced quite completely, and experiments are described quite well. The interpretation of experimental data agrees with current literature, although discussion is limited. Conclusions are sound.

The paper can be published, but some minor issues need to be fixed:

·      Introduction, line 32: state of the art SOFC based on YSZ electrolyte are operated at750 -800 °C. The reported T of 1000 °C is not updated.

·      Introduction, line 41: Ce4+ reduction implies increase of electronic conductivity as well as of electrical conductivity, i.e. ionic+electronic. Maybe authors would mention lowering of open circuit potential, instead.

·      Experiment, lines 95-95: electrodes are cured at lower T than maximum measuring temperature. Ag layer might further sinter during testing, introducing some change in the impedance spectra due to the electrode instead of the sample. Authors are invited to comment on this.

·      Results and Discussion, Figure 6. Scales on X and Y axes should be the same. This doesn’t seem to be for Nyquist plots reported.

·      Results and Discussion, line 151: the statement “the polarization resistance value is very small, so the corresponding semi-arc is not formed on the AC impedance curve” is not clear, because the quantity “polarization resistance” is typically limited to electrode losses as sum of ohmic, charge transfer and diffusion phenomena. In this case the subject is the electrolyte, where especially at high temperature there are no “polarization” effects.

·      Results and Discussion, line 192: figure 9 reports the thermal expansion vs. T, not the TEC, which is the slope of it. In addition, linearity look limited to 400-1000 °C range. Authors are invited to rephrase line 192, for example as “..thermal expansion of CSYG-15%BIT07 shows linear in the range 400-1000 °C”.

A general revision of English would also be beneficial.

Reviewer 3 Report

The paper presents the material study of CeO2-BaTiO3 based ceramic composites for IT-SOFC electrolyte. The authors demonstrate the material development, characterization and electrochemical assessment to verify the potential of the composites as IT-SOFC electrolyte. The idea is interesting and the attempt to investigate the material from various aspect is proper to study such a compound, however, there are many doubtful descriptions and unfortunately the result would not convince that the composite is promising for the IT-SOFC electrolyte. Significant revisions are required to be considered this for publication in this journal. The comments and questions are in the following.

1. This is just a comment, but I think it is not anymore common to operate YSZ based SOFC at 1000°C but the operational temperature has been reduced even in YSZ electrolyte supported cells, since there have been a lot of effort and development of the materials and techniques.

2. The XRD patterns do not match with what the authors explain. There are obvious changes in the diffraction pattern of BIT07 phase with different BIT07 ratio (most significant is around 30 degree). As far as I know, the composition of BIT07 has only one peak around 30 degree. There seem more peaks in some samples. It is also notable that the CSYG diffraction angles are shifted with BIT07 ratio. If there is no reaction between these two phases and they are chemically compatible as the authors claim, peak shifts or diffraction pattern changes should not be observed. The authors should analyze the results more carefully.

 3. About the FT-IR spectra, the authors should provide the information of the FT-IR measurements, in which mode and on which type of sample form they were performed. Since the samples are powder or ceramics, these would be important information to understand the data. I also wonder why the authors are dealing with -OC, or COO-, as well as NO3 components. Is Bi-O a typo of Ba-O in page 4, line 118? Are the FT-IR specimens not sintered? As most of the oxide compounds show O-H vibration, O-H stretching peaks are not surprising, but I wonder if there are carbonates left in the specimen even after sintering? There are too many questions for this issue. What can we understand from the result in this case?

 4. The quality of SEM images in Figure 5 is low. Was the specimen polished on the surface for SEM observation? The scale for each sample needs to be clear. It is difficult to qualify the sample in this figure.

 5. TEC values from the literatures are wrong. LSM, LSCF are not so small.

If the obtained TEC value of 9x10-6 is correct, considering the correct TEC values, LSM ~ 12 x10-6, LSCF ~ 14 x10-6 or larger, the small TEC of CSYG-15%BIT07 composite is not suitable as an electrolyte material with LSM or LSCF electrode.

6.  Page 5 line 145, it is written that two semicircles appear at 500°C while there are no semicircles above 600-800°C. I do not understand which part are the authors claim at 500° and at the other temperatures. There seem two or more semicircles above 600°C, too.

7. Why conductivity is plotted in a linear scale in figure 7? It is hardly compared among the samples at low temperatures in this plot. If figure 7 is to demonstrate the conductivity vs composition, it is better to plot the conductivity at only a few temperatures versus BIT ratio.

The Arrhenius plot in figure 8 demonstrates that the conductivity data are not always linear in this expression, which suggests that either there could be a change of the conduction phenomena at some temperature or the data are scattered. It is hard to judge if the activation energy calculation is reasonable from figure 8.

8. Figure 10 should be corrected. Firstly, the power at 800°C should not be 1 W/cm2. Either the current density data or the calculation of power is wrong. From the plot, it looks like that the performance is around 0.15 A/cm2. If this is correct, this is not very high in comparison to the state-of-the-art electrolyte materials. I could not find any description about the sample dimension (size, electrolyte thickness), electrode area, which is concerned when the performance is compared. Please provide information about them. Secondly, the plot of the performance between 600 and 750°C are wrong. They don’t fit with the cell voltage profile.

9. English should be improved for the entire text.

Reviewer 4 Report

Dear Authors, thank you for your article, submitted to Materials. I have read it with interest, because studied composite materials may be really considered as prospective electrodes for intermediate-temperature SOFCs. However, some moments must be corrected before the publication, because I have the following serious comments on your manuscript:

1. In my opinion, References [1-4] at Line 27 are out of context, because they are concerning doped ceria, not SOFCs - as in the text.

2. Line 30. The choice of References [3,5] is not the best in context. Probably, there are more relevant references- devoted to YSZ specifically, not to materials for SOFCs in general. 

3. It is not clear from Intoduction Part about the correllation between Ba2In2O5 and In-doped BaTiO3.

4. Sample Preparation. Please, add the analytical grade of initial reagents.

5. Experimental Part does not contain the description of IR spectroscopy experiments. Also, methodology of cells' electrochemical characterization is absent. 

6. From Figure 3, one can see, that XRD patterns of the 10BIT07, 15BIT07 and 25BIT07 samples have additional peaks, not corresponding to doped ceria and BIT both. What are they? Please, add XRD patterns of basic materials: doped ceria and BIT.

7. Lines 115, 116. It is not clear, why were bands attributed to O-H, -OC, COO- strchings, and NO3-ion for the sample sintered at 1300 oC? Line 118. Why were Bi-O bonds in samples without Bi?

8. Conductivity measurements. What is the nature of charge carriers in the studied composite material? Please, add the data for basic compounds for approving the high ionic conductivity of the composite materials.

9. Cells' characterization. It is nesessary to provide the experiments of the chemical compatibility between electrodes and electrolyte and electron microscopy data for electrode/electrolyte layers, that can approve the absence of chemical interactions at phase boundaries and cells' defects also.

10. It is wishable to provide the long-term perfomance for the studied single cell.

Author Response

Dear  reviewer,

We would like to thank you for giving us a chance to resubmit the paper, and also thank the reviewers for giving us constructive suggestions which would help us both in English and in depth to improve the quality of the paper.

Thank you very much!

Sincerely yours,

Changan Tian

Round 2

Reviewer 1 Report

In this work, the Ce0.8Sm0.16Y0.03Gd0.01O1.9 and BaIn0.3Ti0.7O2.85 composites with various ratios have been prepared and evaluated as new electrolytes for solid oxide fuel cells. The crystal structure properties, Density and Microstructure, electrical conductivity, thermal expansion and electrochemical properties in SOFC are investigated. Unfortunately, the work cannot be published without significant improvements. The paper in present form in my opinion still does not meet the requirement of the journal of Materials. However, the work can be resubmitted after significant improvements. In the following, there are some suggestions for improvement which can be considered in a future submission. 

1.For this work, extensive editing of English language and style is mandatory, such as “Solid oxide fuel cells (SOFCs) have the advantages of high energy conversion efficiency, low pollution, low cost and environmental friendliness, and is considered as a promising energy conversion device [1-3]. Electrolyte, a key component of SOFCs, act as conducting ions, isolating fuels and oxidizing gases.”

2. The introduction part should be optimized to justify the study. 

3. All abbreviations, such as: LSM, LSCF, LSM-YSZ should be explained.  

4. In the part of “3.1. Structural Analysis”, the reviewer can not understand the sentence of “And for the composite electrolytes of 5%BIT07, 10%BIT07, 120 15%BIT07, 20%BIT07 and 25%BIT07, not only LSGM and GDC,but also new other phase 121 can be identified from their XRD patterns.” Where do the LSGM and GDC phases come from?

5. In the section of “3.5. Single Cell Performance”, Figure 10. “I-V/I-P Characteristics of fuel cell at 800 °C” should be corrected. Only one temperature (800C) measurement is far from good. V-I data with impedance data in function of temperature are of great interest for the readers of the journal of Materials.   

Reviewer 3 Report

The authors have revised some of the issues pointed out in the 1st review process, which has improved the manuscript to some extent, however, there are still several critical points to be published in a journal. The comments and questions are in the following.

1.      After all, which temperature range do the authors consider for IT-SOFC? From the revision, it seems the authors are concerning only near 800°C, but this is then not competitive with the operational temperature of YSZ based SOFC near 800°C. The temperature of 800°C would be the highest for IT-SOFC, which could start from around 600°C usually. The lowest activation energy of 0.89 eV obtained in this work is not very low and could decrease the performance at moderate temperatures. If any, the other advantage has to be claimed in the manuscript. E.g. lower sintering temperature, no need of Ceria protection layer, etc. It is also not described in the introduction why the authors use three dopants in CeO2. It is important to explain the reason and why they decided to use this specific composition, and if any, it is better to describe the superiority of this composition to SoA ceria electrolytes. Those would improve the quality of this paper.

2.      The authors have revised the text for XRD data. The text is now more appropriate and it is noted that the two materials react at 1300°C. However, there is no discussion or more information other than “new other phase” and peak shifts of the two phases, indicating the change of the components. It is critical to consider the secondary phase and the phase shifts of the electrolyte, if they have no significant impact in the fuel cell application.

3.      About the FT-IR spectra, though the authors provided more information of the FT-IR measurements in the experimental section, the questions on FT-IR results remain.

a.      Why the authors are dealing with C=O and C-H. Why there are carbonates left in the specimen even after sintering? Are they only on the surface of the sample powder? It could happen when any excess BaO is formed.

b.      Is Bi-O a typo of Ba-O in page 4, line 134? If not, could the authors explain where the Bi came from?

c.      After all, what could the authors conclude from FT-IR? I do not see anything meaningful from the results in terms of the issue the authors concluded in this section. There seem more problems rather than valuable information. I have to stress here that only Ce-O and Ba-O (if Bi-O is a typo of this) vibration bands do not guarantee the formation of CSYG and BIT07. They could also be of other phases.

4.      The low quality of SEM images in Figure 5 cannot be improved by merely enhancing its size. I wonder if the Fig.5 d is useful in addition to Fig.5 c (and I think the 1500 in the figure caption is the typo of 15000). The questions given in the last review remain: Was the specimen polished on the surface for SEM observation? The scale for each sample needs to be clear. It is difficult to qualify the sample in these pictures for that purpose.

5.      It is good that the authors have corrected the TEC values from the literatures.

From the mechanical point of view, the TEC value of 9.13x10-6 is not moderately low, but already in a critical range for SOFCs with LSCF, LSM or Ni-cermet. If their intension is to match the TEC of this composite with LSM-YSZ and SrTiO3 based electrodes, it could make sense though.

6.      Though the text for Fig. 10 is revised to correct the power density at 800°C, the number in the figure is still wrong. Please correct the figure. I wonder why the authors deleted the data of the other temperatures. It would be better to include the performance data of other temperatures in this plot.

7.      Unfortunately, there are still many grammatical errors in English. The text should be checked more carefully and revised.

Reviewer 4 Report

Dear Authors, thank you for your attention to my questions and comments! After the reading the revised manuscript, I would like to propose the following remarks:

1. Line 232. The comparison of literature data for maximum power output  is nesessary to perform at the same temperatures. It is wishable, at 700 oC - in the middle of the intermediate-temperature working range of IT-SOFCs.

2.  BaIn0.3Ti0.7O2.85 is In-doped derivative of BaTiO3 with perovskite structure. Ba2In2O5 has the brownmillerite structure below 930 oC (but only above 930 oC the disordered oxide Ba2In2O5 is characterized by the increase in oxygen-ion conductivity). Please, provide the transition between perovskite and brownmillerite structures due to doping with using the defect models. It will help to clearify the main aim of investigation -  to create the newest electrolyte materials with high level od ion conductivity.

3. Line 134. Is really Bi (bismuth) there, and in Ref. [26]? In (Ce,Sm,Gd,Y)O-(Ba,In,Ti)O complex oxides?

4. The SEM investigations of the tested cell is needed to approve the absence of cells delamination and materials reactivity.

With Regards, Reviewer

Round 3

Reviewer 1 Report

 Accept in present form

Author Response

Dear reviewer,

We would like to thank you for giving us a chance to resubmit the paper, and also thank the reviewers for giving us constructive suggestions which would help us both in English and in depth to improve the quality of the paper. Here we submit a new version of our manuscript with the title “Preparation and properties of Ce0.8Sm0.16Y0.03Gd0.01O1.9-BaIn0.3Ti0.7O2.85 composite electrolyte”, which has been modified according to the reviewers’ suggestions. A native English-speaking colleagues have been invited to check and improve the English of the manuscript.

Thank you very much!

Please contact me by E-mail:[email protected] if there is any question.

Sincerely yours,

Changan Tian

Reviewer 3 Report

In this version, it is noticed much more revisions are made. The English has much less flaws than the former ones. However, there are still a few critical points to be revised if this is going to be published in this journal. One of the critical issues is that the optimal sample, CSYG-15%BIT07, contains considerable amount of unknown secondary phase observed by XRD. The SOFC electrolytes have been intensively studied since decades to identify several promising materials, which have been tested in more practical cells and stacks for long term operation to demonstrate promising performance and durability. The impact and/or the role of the unknown secondary phase are not discussed at all in the manuscript, while the short-term performance is rather low in comparison to the SoA SOFC. It is not easy to validate their composite potential for SOFC electrolyte. The comments are in the following, with which I would strongly recommend to revise the manuscript further.

1. The authors have revised the text for XRD data further. I have to note here again unfortunately, there is no discussion or more information about the “new other phase” other than its indication. It is crucial since the most important composition, CSYG-15%BIT07 does not show any BIT07 phase but considerable amounts of secondary phases. One should critically consider about such a secondary phase and the phase shifts of the electrolyte, if they would have no significant impact in the fuel cell application. Issues on XRD results are listed, which I would like the authors to reconsider and revise the manuscript.

a.       In the revised version, the authors claim that the CSYG diffraction peak shifts are due to Ba2+ and In3+, which might increase the number of oxygen vacancies in CSYG. Ba2+ is not likely since its ionic radius of 1.42 A in fluorite structure (coordination number: 8) is way larger than that of Ce4+ (0.97 A). To discuss about the lattice parameter change, appropriate information should be given. It is difficult to know how much was the change in this phase.

b.       In the case of 15%BIT07, which is the most important composition in this paper, there are several peaks which might be of a few phases, while there is no BIT07 peak in CSYG-15%BIT07. From the diffraction angles, it looks that there are BaTiO3 (~32°, ~38°, and ~45°) and Ba2In2O5 (~30°, ~53°) This fact indicates the decomposition of BIT07, and I am afraid it is not appropriate to call this sample as CSYG-BIT07 composite. It is possible to identify such phases by a simple analysis, and it is important to explain the analytical results in the manuscript in such a case. Please provide more quantitative information and discuss about the phases with respect to the SOFC application.

2. It is now clear what was observed by FTIR. I am afraid the FTIR results would not bring any additional value to those of XRD, considering the measurement condition (powder specimen: mentioned only in the authors’ responses to the reviewer) and the observed modes. The discussion of this section is rather misleading. I strongly suggest the authors to omit the section of FTIR. It would improve quality of this work.  

3. This is a comment, since I am not able to expect any more revision by the authors unfortunately. The SEM pictures of the unpolished surface is not suitable for the purpose the authors are dealing with in the manuscript. It is often different between the surface and inner part of the sintered ceramics. It would be much clearer image without any influence of the surface condition, when the SEM pictures are made on a cross section of the specimen (polished) to observe the grains and the grain boundaries. 

4. The English has been significantly improved. I still recommend the authors to check the whole manuscript carefully.

e.g. p1, line 30, “YSZ needs to operate at high temperature…”

... could be "YSZ needs to be operated at high temperatures"

Reviewer 4 Report

Dear Authors, thank you for your attention to my comments! I am seeing, that you have improved the manuscript strongly. In particularly, it concerns Introduction Part and Discussion of XRD data.

However, in my opinion, before the articles' publication additional experiments are needed for the approving of the absence of the chemical reactivity between electrodes and electrolytes of the tested single cell: scanning electron microscopy or chemical compatibility investigations.

Besides, values of cells' maximum power density from literature in comparison with the present investigation  (lines 277-282) must be provided for 700 oC - not at 600 and 800 oC randomly.
